# A Preclinical Evaluation towards the Clinical Application of Oxygen Consumption Measurement by CERMs by a Mouse Chimera Model

**DOI:** 10.3390/ijms20225650

**Published:** 2019-11-12

**Authors:** Takashi Kuno, Masahito Tachibana, Ayako Fujimine-Sato, Misaki Fue, Keiko Higashi, Aiko Takahashi, Hiroki Kurosawa, Keisuke Nishio, Naomi Shiga, Zen Watanabe, Nobuo Yaegashi

**Affiliations:** 1Department of Obstetrics and Gynecology, Tohoku University Hospital, Sendai 980-8574, Japan; takashikuno@med.tohoku.ac.jp (T.K.); demic0571@gmail.com (A.F.-S.); fue@med.tohoku.ac.jp (M.F.); keiko.higashi@med.tohoku.ac.jp (K.H.); aiko-takahashi@med.tohoku.ac.jp (A.T.); naomit@med.tohoku.ac.jp (N.S.); zenw5261@med.tohoku.ac.jp (Z.W.); yaegashi@med.tohoku.ac.jp (N.Y.); 2Department of Obstetrics and Gynecology, Tohoku Medical and pharmaceutical university, Wakabayashi hospital, Sendai 984-8560, Japan; acidman99equal831@yahoo.co.jp; 3Institute for Animal Experimentation, Tohoku University Graduate School of Medicine, Sendai 980-8575, Japan; k-nishio@med.tohoku.ac.jp

**Keywords:** assisted reproductive technology, embryo evaluation, chimera, oxygen consumption rate, adenosine triphosphate (ATP), mitochondria

## Abstract

We have developed an automated device for the measurement of oxygen consumption rate (OCR) called Chip-sensing Embryo Respiratory Measurement system (CERMs). To verify the safety and the significance of the OCR measurement by CERMs, we conducted comprehensive tests using a mouse model prior to clinical trials in a human in vitro fertilization (IVF) program. Embryo transfer revealed that the OCR measured by CERMs did not compromise the full-term development of mice or their future fertility, and was positively correlated with adenosine triphosphate (ATP) production and the mitochondrial membrane potential (ΔΨm), thereby indirectly reflecting mitochondrial oxidative phosphorylation (OXPHOS) activity. We demonstrated that the OCR is independent of embryo morphology (the size) and number of mitochondria (mitochondrial DNA copy number). The OCR correlated with the total cell numbers, whereas the inner cell mass (ICM) cell numbers and the fetal developmental rate were not. Thus, the OCR may serve as an indicator of the numbers of trophectoderm (TE) cells, rather than number or quality of ICM cells. However, implantation ability was neither correlated with the OCR, nor the embryo size in this model. This can probably be attributed to the limitation that chimeric embryos contain non-physiological high TE cells counts that are beneficial for implantation. CERMs can be safely employed in clinical IVF owing to it being a safe, highly effective, non-invasive, accurate, and quantitative tool for OCR measurement. Utilization of CERMs for clinical testing of human embryos would provide further insights into the nature of oxidative metabolism and embryonic viability.

## 1. Introduction

In the past, the success of in vitro fertilization (IVF) was contingent on the numbers of embryos to be transferred. An inadequate number of embryos transferred resulted in multiple pregnancies, which posed significant maternal health risks and caused neonatal complications. For this reason, elective single embryo transfer (eSET) is now a widely accepted practice for infertile women of a certain age range, even when several high-quality embryos are available [1,2,3]. Although previous efforts have improved assisted reproductive technology (ART) outcomes, such as the change from transabdominal to transvaginal ultrasound guidance during embryo transfer and an innovation of single medium development [4,5,6], the selection of a high-quality embryo for transfer is justified to increase the chance of pregnancy and reduce the risk of miscarriage.

Various methods have been proposed for modern ART to evaluate embryo quality. Morphological evaluation remains the standard global practice. The embryos are observed under a stereomicroscope at a specific time point and used to predict implantation and pregnancy rates [7,8,9,10]. Several reports have shown that the scoring system based on the morphological features of the embryo are correlated with its developmental competence [11,12]. However, morphological evaluation is not quantitative and is inevitably influenced by inter- and intra-observer variability [13]. Recent technological advances have resulted in a quality assessment method for ART using spent embryo culture medium, cell cycle timing, morphokinetic/morphometric parameter evaluation by time-lapse monitoring (TLM), and preimplantation genetic testing for aneuploidy (PGT-A) [14,15,16]. The aforementioned techniques have provided important new insights into early embryonic development and the selection criteria for developmentally competent embryos. However, none of them are used in the standalone evaluation of the fate of the in vitro preimplantation stage embryo from implantation to live birth.

An important contributor to normal embryo development is consistent metabolism. Mammalian oocytes have excessive mitochondria in their large cytoplasm, which play important roles in oocyte maturation and early post-fertilization embryonic development. In contrast, mitochondrial biogenesis is arrested during early preimplantation embryo development until blastulation [17]. Mitochondria are essential for basic cell functions and cell division because of their crucial roles in metabolic energy production. The primary role of mitochondria is aerobic respiration and the production of metabolic energy in the form of adenosine triphosphate (ATP). Therefore, the oxygen consumption rate (OCR) has recently attracted attention as a metabolic indicator reflecting mitochondrial activity in oxidative phosphorylation (OXPHOS) [18]. We recently assessed embryo respiration as a marker for the selection of viable embryos using a novel device called the Chip-sensing Embryo Respiration Monitoring system (CERMs) [19]. We demonstrated that the CERMs automatically measured the OCR of spheroids, bovine embryos, and frozen–thawed human embryos. The OCR of ≤ 5 embryos could be measured simultaneously by simply placing them on the center of the chip (Figure 1). The CERMs was as sensitive as scanning electrochemical microscopy (SECM) [20,21]. The OCR of developed frozen–thawed human blastocysts was linearly correlated with the multiplicative blastocyst quality score (BQS) (*R^2^* = 0.6537; *p* = 0.008) [19,22]. However, some of these blastocysts presented with a discrepancy between the OCR and the BQS. Therefore, it is postulated that human embryo morphology and metabolism may not be consistently correlated. Moreover, the association between the OCR measured by CERMs and the biological parameters related to mitochondrial activity, embryo viability, and implantation ability have not yet been evaluated. In addition, the effects of OCR measurement by CERMs on in vivo embryo development and the progeny’s future fertility were not evaluated. To warrant a clinical trial on CERMs-based OCR measurement involving embryo transfer, these aforementioned information gaps must first be filled.

In the present study, we used a mouse model in the attempt to answer these questions raised in previous research. In order to verify the full-term development of embryos after the OCR measurement, we measured the OCR of embryos, then transferred them. Subsequent fertility of the adults developing from these embryos was evaluated based on their progeny using a mouse model. Further, we investigated whether the OCR measured by CERMs correlates with known mitochondrial activity markers. We also assessed the relationship between the OCR and the ability of implantation (initiation of pregnancy).

## 2. Results

### 2.1. Effect of Oxygen Consumption Rate (OCR) Measurement by the Chip-Sensing Embryo Respiration Monitoring System (CERMs) on Embryo Development and Their Future Fertility

We first confirmed the safety of CERMs. Three replicated blastocyst transfers, developed from a single mouse embryo, were performed after OCR measurement by CERMs. Six, two, and seven healthy pups were born via embryo transfer, respectively. We then assessed the fertility of the adult mice arising from these embryos and they produced F2 pups. Thus, the OCR measurement by CERMs does not harm in vivo embryo development and their future fertility.

### 2.2. Correlations between the OCR and the Adenosine Triphosphate (ATP) Levels and Cell Counts in Blastocysts Developed from a Single Embryo

Previous studies have demonstrated a correlation between the OCR and the ATP content [20,21]. The blastocyst cell number is correlated with the initiation of pregnancy and, by extension, embryo viability [24]. We assessed the correlations between the OCR and the cell numbers and the ATP levels at the blastocyst stage. Fifty-three blastocysts developed from single embryos and their ATP levels were evaluated after the OCR measurement. The mean ATP and mean OCR were 0.601 ± 0.18 pmol/oocyte and 4.82 ± 2.17 fmol/s, respectively. Contrary to expectation, the ATP level was poorly correlated with the OCR (*r* = −0.247; *p* = 0.075) (Figure 2A). We then determined the correlation between the OCR and the cell numbers. The OCR was measured for 40 blastocysts (mean OCR: 4.10 ± 1.24 fmol/s). The OCR was not statistically correlated with either total (*r* = 0.069; *p* > 0.05) or inner cell mass (ICM) cell number (*r* = −0.091; *p* > 0.05) (Figure 2B,C). These results contradicted our expectations. Therefore, we speculated that either the size or the OCR activity of the mouse blastocysts might be incompatible with CERMs, which were originally designed according to the size of human embryos.

### 2.3. Correlations between the OCR and the ATP Levels and Cell Counts in Chimera Blastocysts

We formed 15 large mouse chimeric blastocysts from the aggregation of pairs of two-cell embryos (Figure 3A). We also generated 18 control blastocysts from single embryos. The mean diameter, average OCR, and ATP level for the single and chimera blastocyst groups were 130 ± 18.2 μm, 3.97 ± 1.55 fmol/s, and 0.602 ± 0.17 pmol/oocyte and 148 ± 18.9 μm, 7.77 ± 3.02 fmol/s, and 1.16 ± 0.28 pmol/oocyte, respectively. The aforementioned parameters were significantly greater in the chimera blastocysts than the control single blastocysts (*p* < 0.05). The OCR of viable human embryos was 5.3 (4.2–6.4) fmol/s and that of 7–9-cell cleavage-stage embryos (D2) and morula-to-blastocyst-stage embryos (D5) was 8.3 (7.3–9.0) fmol/s [19]. Therefore, the chimera blastocyst seemed to be a more appropriate animal model to measure the OCR by CERMs.

Fifteen chimera blastocysts were created and their ATP levels determined following OCR measurement. The mean ATP and OCR for all chimera blastocysts were 1.16 ± 0.28 pmol/oocyte and 7.77 ± 3.02 fmol/s, respectively. The ATP levels were significantly positively correlated with the OCR measured by CERMs (*r* = 0.533; *p* < 0.05) (Figure 3B). We then generated 22 chimera blastocysts and fixed them after the OCR measurements. Chimeric blastocysts with known OCRs were immunohistochemically stained, and the ICM cells and total cells were counted according to the cell counting for blastocyst from a single embryo. The average ICM and total cell counts were 8.18 ± 2.08 and 45.0 ± 11.7, respectively. These were significantly higher than those for single blastocysts (5.15 ± 2.00 and 40.4 ± 10.7, respectively). The total cell number was also significantly positively correlated with the OCR (*r* = 0.523; *p* < 0.05), whereas the ICM was not (*r* = −0.088; *p* = 0.70) (Figure 3C,D).

The BQS of some frozen–thawed human blastocysts were poorly correlated with the OCR [19]. Therefore, we also explored the correlation between morphology and the OCR. No suitable morphological scoring system has yet been established for mice. For this reason, we regarded blastocyst size as a morphological parameter. We investigated the correlation between size (mean diameter) and the OCR for 15 chimera blastocysts and found that the mean diameter was poorly correlated with the OCR (*r* = −0.144; *p* = 0.61) (Figure 3E). CERMs requires embryos of the appropriate size and/or respiratory activity to be able to evaluate embryo activity accurately. A mouse chimeric blastocyst may, therefore, be a suitable model to assess the impact of the OCR measurement on embryonic development by CERMs.

### 2.4. Correlation between the OCR and Mitochondrial Activity

Mitochondrial DNA (mtDNA) replication does not occur in the cleavage stage until peri-implantation. Consequently, mtDNA in the unfertilized oocyte is critical for subsequent embryonic development [25]. Mitochondrial membrane potential (ΔΨm) is closely correlated with ATP production and, by extension, mitochondrial OXPHOS activity and post-fertilization embryonic development [26,27]. We first performed real-time PCR to quantify the mtDNA copy number for 16 chimera blastocysts of known OCR. However, the OCR was not correlated with 2-ΔΔCt (*r* = 0.099; *p* = 0.72; Figure 4A).

We ran a mitochondrial membrane potential (ΔΨm) assay using JC-10. Thirteen chimera embryos were generated and their OCRs were measured. The ratios of red fluorescence (J-aggregate) to green fluorescence (monomeric form) were determined for the embryos with the three highest and three lowest OCRs (Figure 4B). The mean red/green ΔΨm (R1/G1) for the top three (mean OCR: 20.73 fmol/s) and the bottom three (mean OCR: 9.23 fmol/s) embryos were 0.11 and 0.05, respectively. Unlike the mtDNA copy number, the mitochondrial membrane potential increased with the OCR, albeit the sample size is small.

The OCR did not reflect the mtDNA copy number, which indicates the numbers of mitochondria in the blastocyst. However, the mitochondrial membrane potential tended to increase with the OCR. This finding indirectly proves that the OCR, ATP level, and mitochondrial membrane potential are positively correlated during embryo development.

### 2.5. The OCR and In Vivo Embryo Viability

We analyzed the relationship between the OCR and the ability of implantation. Seven replicated blastocyst transfers were performed after the OCR measurements and implantation sites were assessed at 12.5 dpc (Figure 5A,B and Appendix A). Twenty chimeric blastocysts were segregated into two groups according to their OCR (10 high and 10 low). They were transferred into each side of the bicornuate uteri of recipient ICR mice. One of the experiments was excluded because technical failure occurred in one of the transfers and no implantation was observed in one side of the bicornuate uterus. The implantation rates for the high-OCR group (mean OCR: 14.71 ± 4.61 fmol/s) and the low-OCR group (mean OCR: 7.65 ± 1.50 fmol/s) were 34/60 (56.7%) and 42/60 (70.0%), respectively (*p* = 0.27) (Table 1). The numbers of fetuses per embryo for the high- and low-OCR groups were 10/60 (16.7%) and 23/60 (38.3%), respectively (*p* = 0.85) (Table 1). The weights of the fetuses for the high- and low-OCR groups were 80.93 ± 26.4 g and 90.37 ± 28.5 g, respectively (*p* = 0.39) (Table 1).

The OCR was neither correlated with implantation ability, nor fetal development in mouse chimera blastocysts in vivo. Therefore, we conducted four replicated embryo transfers based on blastocyst morphology (size) (Appendix A). One of the experiments was excluded because of technical failure. The implantation rates for the large blastocyst group (mean OCR: 9.28 ± 3.01 fmol/s) and the small blastocyst group (mean OCR: 10.2 ± 3.96 fmol/s) were 20/30 (66.7%) and 17/30 (56.7%), respectively (*p* > 0.05) (Table 2). The morphological types did not significantly differ in terms of number of fetuses per implantation site or average fetal weight (*p* > 0.05) (Table 2).

The OCR of the chimera blastocyst was not correlated with implantation ability or fetal development. Moreover, chimera blastocyst size was not correlated with implantation ability.

## 3. Discussion

In the present study, we demonstrated that the OCR measurement by CERMs does not harm embryo development in vivo or progeny fertility. However, normal mouse blastocysts, developed from a single embryo, did not correlate with the cell number or the ATP level, possibly because the size and/or respiratory activity of mouse single blastocysts were incompatible with the chip-sensor originally designed for use with human embryos. Utilizing chimeric embryos, however, the OCR was correlated with both the cell number and ATP content, but not with morphology (size). Mitochondrial membrane potential tended to increase with OCR, whereas mtDNA copy number did not. Neither the OCR, nor morphology (size) of mouse chimeric blastocysts were correlated with implantation ability.

Establishing an animal model is indispensable for conducting invasive assays, assessing the full-term development of embryos and the fertility of the progeny, and verifying the impact of OCR measurements before starting actual clinical trials. The current version of CERMs involves exposing embryos to atmospheric air and HEPES-buffered measurement media. Further, the principle of the OCR measurement requires the application of an electric potential to embryos to be measured. Drastic environmental changes may have a negative impact on embryonic development and epigenetic modifications [6]. Therefore, these effects of environmental changes on embryonic and progeny development may be of concern. Indeed, a single study in mice demonstrated that the OCR measurement somewhat compromised blastocyst development, albeit the OCR in cleavage stage embryos may be an indicator (though a not a strong one) of blastocyst development [28]. In the present study, we demonstrated that the OCR measurement by CERMs does not harm embryonic development in vivo or the fertility of the progeny. Thus, CERMs can be safely employed in future clinical trials in IVF. Although an adjustment of the diffusion coefficient is required, CERMs can be upgraded and incorporated into incubators. Thus, automatic OCR measurement can be performed under 5% CO_2_ in the embryo culture media in the near future, so that oxidative stress and drastic environmental change can be controlled for.

Our initial attempts to measure the OCR in mouse blastocysts derived from single embryos revealed no reasonable correlations between the OCR and the ATP level or the cell count. These findings suggest that CERMs is not universally applicable for all mammalian embryo sizes and respiratory activity levels. The size and the distance of the microelectrodes encircling the pit in the chip-sensor are determine based on the size of human embryos and respiration rates. Our previous, study revealed that the sensitivity of CERMs was as comparable to that of SECM when the appropriate embryo or cell size was used [19]. The OCR of up to five embryos could be simultaneously measured by CERMs, but OCR of embryos with small size and/or low respiratory activity could not be accurately evaluated, due to the limitation of the chip-sensor. In contrast, SECM is a more universal technique, because scanning can be manually adjusted via microelectrode manipulation to suit the requirements of different sizes [20,21]. In this study, we successfully overcame this limitation by using chimeric mouse blastocysts, whose size and respiratory activity were similar to those of human embryos (e.g., mean diameter: 120–250 m; mean OCR: ≥4 fmol/s).

However, if embryo size (morphology) is a major criterion influencing respiratory activity, OCR measurement may not be a substitute for standard morphological evaluation. In this regard, the size of chimeric mouse blastocysts was poorly correlated with the OCR. Therefore, the OCR of embryos is independent of embryo size, i.e., embryo morphology.

The energy metabolism of preimplantation embryos changes through development from cleavage stage to blastocyst stage, where the oxygen consumption also changes depending on the developmental stage of embryo. Previous studies have demonstrated that blastocysts are mainly dependent on “aerobic glycolysis” to produce ATP [29,30,31,32]. This metabolic change may be essential to prepare for the hypoxic environment that the embryo will encounter post-implantation. However, it is a well-known fact that the glycolytic pathway is an inefficient in generating ATP than that of mitochondrial OXPHOS. A previous study on pigs has demonstrated that the overall contribution of “aerobic glycolysis” for ATP production at blastocyst stage is relatively small, and thus it mainly relies on OXPHOS [33]. Other studies have also demonstrated that 30% of oxygen consumed both in mouse and rabbit blastocysts relies on the non-mitochondrial process (non-OXPHOS) [34,35], while mouse cleavage stage embryos consume up to 70% of oxygen by non-OXPHOS [34]. Although both the mechanism responsible for the non-OXPHOS pathway and whether a direct measures of oxygen consumption is considered as a measure of ATP production have yet to be sorely clarified in early embryonic development, embryo substantially increases mitochondrial OXPHOS dependent oxygen consumption and ATP production in blastocyst. Thus, the OCR is closely correlated with mitochondrial activity and ATP production [36]. We therefore used a mouse model and focused on the blastocyst stage, wherein the embryos consume oxygen mainly via the OXPHOS pathway. Since the OCR significantly correlated with the cell numbers and the ATP level in chimera blastocysts, the OCR measured by CERMs can be used as a marker for active mitochondrial metabolism (OXPHOS), ATP production, and cell numbers in blastocyst.

The present study showed that the mtDNA copy number is not correlated with respiratory activity in the mouse blastocyst. Mitochondria play important roles in blastulation, expansion, and hatching [17]. MtDNA replication does not occur during the cleavage stages of mouse embryogenesis [37,38]. The mitochondrial content of the unfertilized oocyte should suffice to maintain vertebrate development up to implantation [25,37]. A reduction in oocyte mitochondria with age has been reported in older females [39,40]. MtDNA copy number may be an indicator of oocyte quality. Higher mitochondrial content or mtDNA copy numbers in oocytes are associated with superior oocyte quality and better clinical outcomes. Nevertheless, recent human studies have revealed that low mitochondrial content in the developing blastocyst may be indicative of its quality and chromosomal integrity [41,42]. High numbers of mtDNA copies may indicate large numbers of mitochondria. However, evaluation of mtDNA copies may be interrupted with care, where a single mitochondrion may contain multiple genome copies. Previous reports have demonstrated that the numbers of mtDNA copies per mitochondrion may vary from 0–15 in human cells [43,44]. Further, the real-time PCR method influences the result of mtDNA copy numbers by either relative or absolute quantification. Given that mitochondrial replication does not occur during preimplantation embryo development up to blastocyst stage, low mtDNA copy number per cell by relative quantification, not in a whole embryo, may be of consequence of a large number of cells in the blastocyst due to segregation of mitochondria with frequent mitosis. In the present study, we utilized relative quantification with copies of nuclear DNA as internal control (the 2-ΔΔCt method). Therefore, mtDNA copy number may not directly reflect mitochondrial density or respiratory activity in blastocysts. A recent study showed that the mtDNA copy numbers in human blastocysts remained constant, irrespective of ploidy, age, or implantation potential [45]. On the other hand, our study showed that both mitochondrial membrane potential (ΔΨm) and the ATP level were correlated with OCR. For this reason, the OCR could serve as a non-invasive, indirect marker of mitochondrial membrane potential (ΔΨm), ATP production, and, by extension, mitochondrial OXPHOS activity, not for the number of mitochondria within the blastocyst [26,27].

Contrary to our expectation, the embryo transfer study showed no change in implantation ability with increasing OCR. However, it has known that the implantation rates of zona pellucida (zona)-free embryos are lower than those for embryos with intact zona [46]. In our research, the average implantation rate of zona-free chimeric blastocysts (62.8%) resembled, or were even higher than, that for zona-intact blastocyst transfer (56.2%; data not shown). On the other hand, the total cell number in the blastocyst is critical for the initiation of pregnancy. A proof of concept was reported for a primate chimera study (three embryos were aggregated and produced a single chimeric embryo), wherein extremely high pregnancy and implantation rates (100% and 71%, respectively) were observed [24]. A hypothesis was formulated, in which non-physiologically high numbers of trophectoderm (TE) cells in mouse chimera blastocysts may have high implantation potential, irrespective of the OCR. Therefore, this model might not be suitable for the assessment of ability of implantation, despite the fact that OCR measurements by CERMs were correlated with mitochondrial activity, which may represent metabolically more active embryos. The initiation of pregnancy was correlated mainly with the TE numbers, whereas the OCR was not correlated with the ICM numbers. Further, the high-OCR group did not show superior fetus rates within the implantation sites. Therefore, we speculated that the OCR may indicate the initiation of pregnancy (TE cell number and quality), rather than embryo proper quality (ICM epiblast) per se and/or chromosomal integrity.

Our study demonstrated that OCR measurement by CERMs may effectively evaluate embryo metabolism, particularly mitochondrial OXPHOS activity, and perhaps TE cell quality and number. Mammalian embryo development is influenced by proper segregation of genetic material, gene expression, cell differentiation, and energy supply. However, it is difficult to assess all of these factors in a single assay. Therefore, the OCR measurement may complement an existing assay, such as chromosomal instability by PGT-A and/or TLM, by revealing metabolic activity, implantation ability, and in vivo embryo development. In this regard, CERMs could be safe and effective due to semi-automated, simultaneous measurements of ≤5 embryos, and is scalable and non-invasive. On the other hand, our study showed no positive correlation between the OCR and implantation ability in the mouse chimera blastocyst. We attributed the results of our embryo transfer study to the mouse model limitation of non-physiological cell numbers in the chimera blastocyst. However, according to the “quiet embryo hypothesis”, overactive embryonic metabolism occurs when the embryos are under stress and/or have reduced developmental potential [47]. Further, the production of ATP via the mitochondrial OXPHOS pathway would inevitably increase the production of reactive oxygen species (ROS) production, called mitochondrial ROS (mROS). High level of mROS, a consequence of the overactive mitochondrial OXPHOS, may have a negative impact on cell structures and macromolecules, such as lipids, membranes, proteins, and DNA, and thereby embryo quality [48,49]. Nevertheless, it remains to be established whether a high OCR, which indicates a high oxidative metabolic rate, may also be indicative of high- or low-quality blastocysts. As CERMs has been found to be a safe, highly effective, non-invasive, accurate, and quantitative tool for OCR measurement, it can be safely employed in studies with human embryos. We believe that CERMs could be a solution for technical issues that have hampered the popularization of OCR measurement for embryos, whereby enhancing study in the field of oxidative metabolism in early embryonic development. Effects related to oxidative metabolism and embryonic viability, such as implantation ability and in vivo embryo development, can be validated in future clinical trials using humans due to the clinical impact of CERMs.

## 4. Materials and Methods

### 4.1. Ethical Approval

All animal procedures were performed in compliance with the Tohoku University protocols and ethical guidelines. Animal experimentation were approved by The Institutional Animal Care and Use Committee of the Tohoku University Environmental & Safety Committee (2015MdA-343-1, 1 Jan 2016).

### 4.2. Embryos and Recipient Mice

Either C57BL6 or B6D2F1 mice were used to create embryos via IVF. ICR mice served as host mothers for the embryo transfer experiment. Eight- to 12-week-old female mice were superovulated by i.p. injection of 10 IU PMSG (ASKA Animal Health, Tokyo, Japan), followed by another i.p. injection of 10 IU hCG (ASKA Animal Health, Tokyo, Japan) 48 h later. Cumulus–oocyte complexes (COCs) were collected from the oviducts at 15–17 h after hCG injection and pre-incubated in TYH medium (LSI Medience Co., Tokyo, Japan) at 37 °C under 5% CO_2_ in air for 60 min until insemination. Sperm was collected from the epididymis of 8- to 12-week-old male mice at the same time as oocyte collection. The sperm was incubated in TYH medium at 37 °C under 5% CO_2_ in air for 60 min until insemination. Three microliters of sperm suspension was added to 200 μL TYH containing COCs. The COCs were cultured at 37 °C under 5% CO_2_ in air for 3 h. The cumulus cells were then washed, denuded, and cultured for 24 h. Two-cell embryos were selected and transferred to KSOM medium (ARK Resource Co., Kumamoto, Japan) and maintained at 37 °C under 5% CO_2_ up to the blastocyst stage. Blastocyst formation and embryo number were visually assessed at 3.5 dpc.

### 4.3. Chimeric Embryo Production

Chimeric blastocysts were generated according to the procedures of Mints and Tarkowski [50,51]. In brief, the zona pellucida of two-cell embryos were manually removed with XYClone^®^ under a red-i 20x laser objective (Hamilton Thorne Ltd., Beverly, MA, USA) at 24 h after insemination (Appendix A). A pair of two-cell embryos was placed in a depression made with an aggregation needle (BLS Ltd., Budapest, Hungary). Fully aggregated chimeric blastocysts were examined at 3.5 dpc (Appendix A).

### 4.4. OCR Measurement by CERMs

OCR measurement was performed according to the method reported in an earlier study [19]. Measurement media was also adapted from a previous study because it is critical to use recommended media for accurate measurement. Briefly, preheated Cook^®^ Sydney IVF Follicle Flushing Buffer (COOK Medical Australia Pty Ltd., Brisbane, Australia) supplemented with 10% Serum Substitute Supplement (SSS^TM^; Irvine Scientific, Santa Ana, CA, USA) served as the measuring medium. The measuring plate contained a chip-sensor and was filled with 2 mL measuring medium without mineral oil then covered to prevent evaporation. Up to five blastocysts were washed with Hepes solution and placed in the pit of the chip with a Pasteur pipette. The measurement program was adapted using a previously published setting for human embryos [19]. Briefly, the potential was held at −0.75 V vs. Pt for 10 s and repeated 3 ×. Three 10-s measurements were conducted at 10-s intervals for a 50-s total measurement time. Each blastocyst was numbered to assess the correlation between the OCR and the other experimental results and to determine its fate at embryo transfer. Measurements were conducted automatically, except for placing and retrieving the sample. The OCR measurement was made outside an incubator. The system was exposed to room air for ≤ 5 min. The sample was retrieved, washed in Hepes solution, and returned to the culture medium until the next experiment. Details of the equipment architecture and measurements principles are reported elsewhere [19].

### 4.5. ATP Measurement

ATP was measured with an IntraCellular ATP kit (TOYO B-Net, Tokyo, Japan), according to the manufacturer’s instruction. Briefly, blastocysts of known OCR were rinsed with phosphate-buffered saline (PBS) and plated on a 96-well plate. Luciferase absorbance was measured by TriStar2 LB942 (Berthold Technologies GmbH, Bad Wildbad, Germany). A standard curve was generated using a 10-fold dilution series of a lysate with a known ATP concentration. The ATP level of each blastocyst of known OCR was then interpolated from the standard curve.

### 4.6. Immunocytochemical Procedure and Cell Count

Immunocytochemical procedures were performed as previously described [52]. In brief, blastocysts were fixed in 4% paraformaldehyde and permeabilized with 0.2% Triton X-100 and 0.1% Tween-20. Nonspecific reactions were blocked with 10% normal serum (Nichirei Biosciences Inc., Tokyo, Japan). Cells were incubated for 40 min in primary antibodies, washed 3×, and exposed to secondary antibodies conjugated with Alexa Fluor (Thermo Fisher Scientific, Waltham, MA, USA) before co-staining with 2 µg/mL of 4’,6-diamidino-2-phenylindole (DAPI: Sigma-Aldrich Corp., St. Louis, MO, USA) for 10 min. Whole cells were mounted on microscope slides and examined by epifluorescence microscopy (Nikon, Tokyo, Japan). Anti-NANOG antibody (ReproCELL Inc., Kanagawa, Japan) was used to detect ICM epiblast (ICM cell number). All blastocyst cell nuclei were stained with DAPI (total cell number). Cells were counted with NIS-Elements (Nikon, Tokyo, Japan). Images were pseudocolored and analyzed in Adobe Photoshop CS2 (Adobe Systems Inc., San Jose, USA).

### 4.7. MtDNA Copy Number Assay

The mtDNA copy number assay was performed with a Mouse Mitochondrial DNA Copy Number Kit (Detroit R&D Inc., Detroit, MI, USA), according to the manufacturer’s instruction. Since little genomic DNA can be extracted from a single blastocyst, whole-genome amplification was performed with a REPLI-g Single Cell Kit (Qiagen, Hilden, Germany) before mtDNA quantitative real-time PCR in a Fast Real-time PCR system (Applied Biosystems, Foster City, CA, USA). Relative quantification was performed by the 2-ΔΔCt method with β-actin as the internal control.

### 4.8. Mitochondrial Membrane Potential Assay

The mitochondrial membrane potential (ΔΨm) was measured with a JC-10^TM^ Mitochondrial Membrane Potential Assay kit (AAT Bioquest Inc., Sunnyvale, CA, USA), according to the manufacturer’s instruction. Briefly, JC-10TM is a dual-emission, potential-sensitive indicator, which preferentially accumulates in the mitochondria. The distributions of JC-10^TM^ monomers (green fluorescence: G1) and J-aggregate fluorescence (red fluorescence: R1) were determined by epifluorescence microscopy (BZ-X700, Keyence, Tokyo, Japan) over a 30-min period following the supplier’s protocol. The images were processed with Advanced Analysis Software BZ-H3A ver. 1.3 (Keyence, Tokyo, Japan). The blastocysts were treated with carbonyl cyanide-4-(trifluoromethoxy)-phenylhydrazone (FCCP; Sigma-Aldrich Corp., St. Louis, MO, USA) for 10 min after the initial images were captured. The images were reanalyzed to measure changes in green (G2) and red (R2) fluorescence. Embryos with G1/G2 < 1 and R1/R2 > 1 were used in the assessments. The embryos with the highest and lowest three values were compared using the ratio of red to green fluorescence.

### 4.9. Embryo Transfer

OCR was measured by CERMs in developed blastocysts, which were then transferred to the bicornuate uterus of pseudopregnant ICR female mice according to previously published methods [46,53]. In brief, ICR females confirmed to be in estrus were exposed to vasectomized males. Pregnancy day 1 (1 day post-coitus) was confirmed from the appearance of a vaginal plug. Blastocysts were transferred at 2.5 dpc. To assess implantation ability, embryos were segregated into two groups based on their OCR or morphology (size). For each embryo transfer cycle, > 30 chimeric embryos were generated by aggregating > 60 two-cell embryos. Embryos having blastocysts with the ten highest and ten lowest OCR were identified and grouped. Each embryo group was transferred to each side of the bicornuate uterus of a recipient ICR female, which was then euthanized at 12.5 dpc and evaluated for the number of implantation sites. Eight replicated blastocyst transfers were performed after the OCR measurements. To confirm full-term development, 10 single blastocysts whose OCR were already measured by CERMs were transferred to both sides of the bicornuate uterus and allowed to grow to full term. Three replicated blastocyst transfers were performed. F1 mice obtained via embryo transfer were weaned after 3 weeks, and subsequently mated to produce F2 pups.

### 4.10. Statistics

Correlations between the oxygen consumption rate and the blastocyst cell number, the ATP level, and the blastocyst size were analyzed by Pearson’s correlation coefficient. Relative differences in the sizes of single and aggregated chimeric blastocysts were evaluated with an unpaired *t*-test. The embryo implantation rates in the high- and low-oxygen consumption rate groups were compared with an unpaired *t*-test. All statistical analyses were performed in R v. 3.3.2 (R Core Team, Vienna, Austria). Statistical significance was set at *p* < 0.05.

## 5. Conclusions

The OCR measured by CERMs was safe and associated with mitochondrial activity and some embryo viability (such as cell numbers) when the appropriate size of embryos was used. Due to non-physiological high numbers of TE cells in chimeric embryos, implantation ability was neither correlated with the OCR, nor embryo size in this model. In this regard, clinical testing with human embryos is desirable to clarify the nature of embryo viability and the OCR.

## Figures and Tables

**Figure 1 ijms-20-05650-f001:**
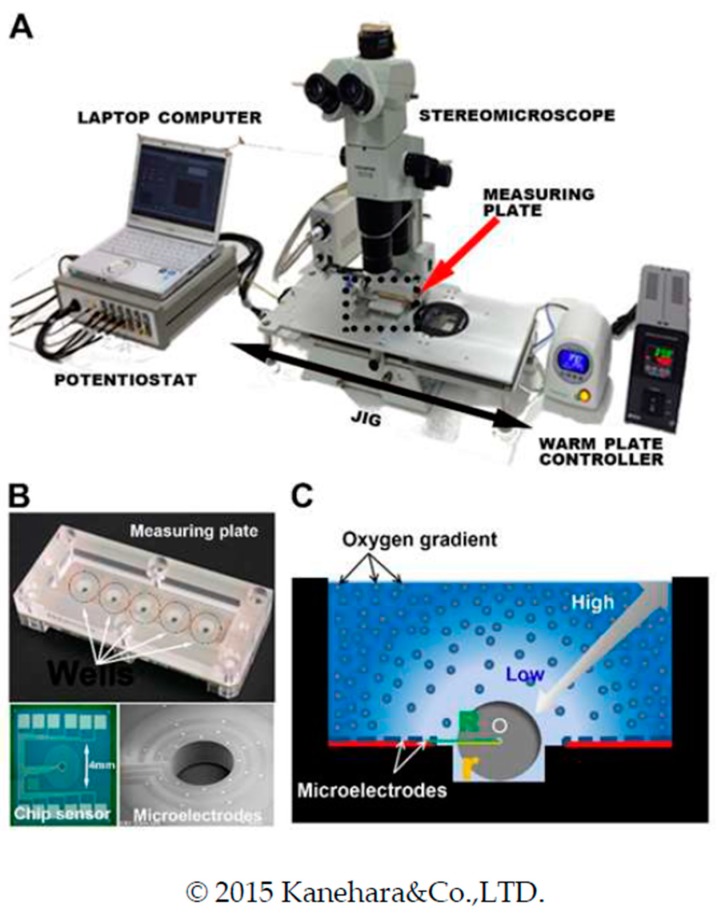
Architecture of the Chip-sensing Embryo Respiration Monitoring system (CERMs). (**A**) Overview image of the architecture of the device, consisting of measuring plate, jig containing built-in warm plate, potentiostat, and laptop computer for analysis. (**B**) The measuring plate consists of five wells, and a chip sensor is implanted in the bottom of the well (top). An enlarged image of the chip sensor in the center of the well is shown (bottom left). Microelectrodes on the chip sensor are arranged in eight different directions, encircling a pit (bottom right). (**C**) Hemispherical area of dissolved oxygen concentration gradient formed by respiration of embryo on the chip. r, radius of embryo; R, distance from center of embryo to electrode. The figure is modified with permission from Shiga et al. [23].

**Figure 2 ijms-20-05650-f002:**
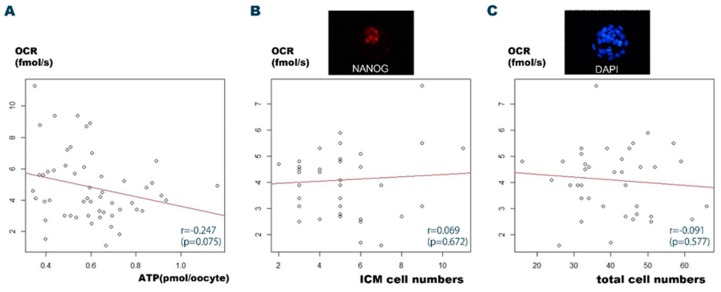
Correlations between the oxygen consumption rate (OCR), the adenosine triphosphate (ATP) levels, and the cell counts of blastocysts developed from a single embryo. (**A**) Scatter plot depicts the correlation between the OCR and the ATP levels in blastocysts developed from a single mouse embryo. The OCR was poorly correlated with the ATP level (*r* = −0.247; *p* = 0.075). (**B**) Scatter plot depicts the correlation between the OCR and the inner cell mass (ICM) numbers in blastocysts developed from a single mouse embryo. The OCR was poorly correlated with the ICM numbers (*r* = 0.069; *p* = 0.672). The small window shows a representative NANOG positive epiblast (ICM) within a mouse blastocyst. (**C**) Scatter plot depicts the correlation between the OCR and the total cell numbers in blastocysts developed from a single mouse embryo. The OCR was poorly correlated with the total cell numbers (*r* = −0.091; *p* = 0.577). The small window shows representative 4’,6-diamidino-2-phenylindole (DAPI)-positive nuclei in the cells of a mouse blastocyst. The original magnification of the immunohistochemical images was 400 ×.

**Figure 3 ijms-20-05650-f003:**
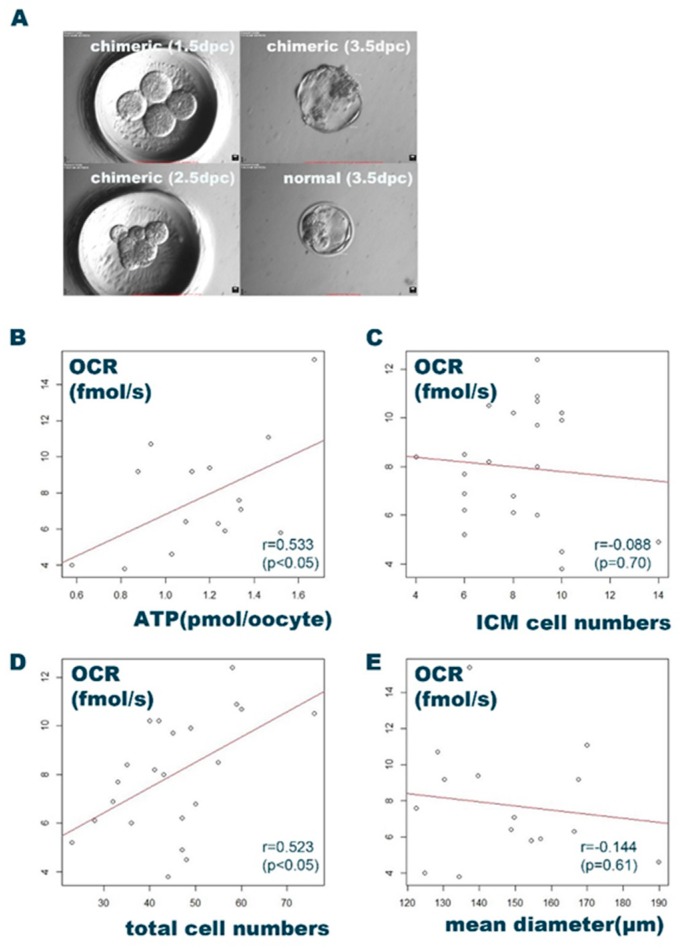
Adapting chimera blastocysts for CERMs evaluation in mice. (**A**) Figures depicting the preparation of mouse chimaera blastocysts by aggregating pairs of two-cell embryos. The chimaera blastocyst (top right) was larger than that developed from a single embryo (bottom right). The original magnification was 300 ×. (**B**) Scatterplot depicting the correlation between the OCR and the ATP level. The ATP level was significantly correlated with the OCR (*r* = 0.533; *p* < 0.05). (**C**,**D**) Scatterplots depicting the correlation between the OCR and the total cell and ICM epiblast numbers in chimera blastocysts. the ICM numbers were not correlated with the OCR (**D**) and total cell numbers were correlated with the OCR (**D**). (**E**) Scatterplot depicting the correlation between the OCR and the morphology (size) of the chimera blastocyst. No correlation was observed between the OCR and the morphology (*r* = −0.144; *p* = 0.61).

**Figure 4 ijms-20-05650-f004:**
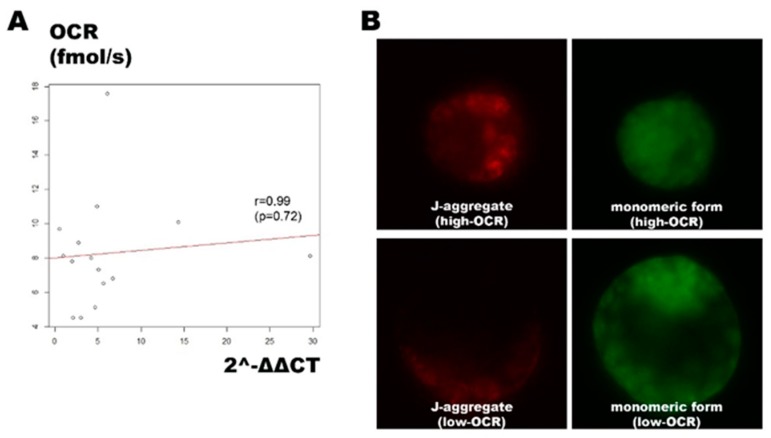
Correlation between the OCR and the mitochondrial DNA (mtDNA) copy number and mitochondrial membrane potential. (**A**) Scatterplot depicting a weak correlation between the OCR and the mtDNA copy number. (**B**) Fluorescent microscopic images of JC-10 staining representing embryos with high-OCR (upper) and low-OCR (lower). Embryo with high-OCR exhibited strong J-aggregate (red) and weak monomeric form (green) signals. The original magnification was 200 ×.

**Figure 5 ijms-20-05650-f005:**
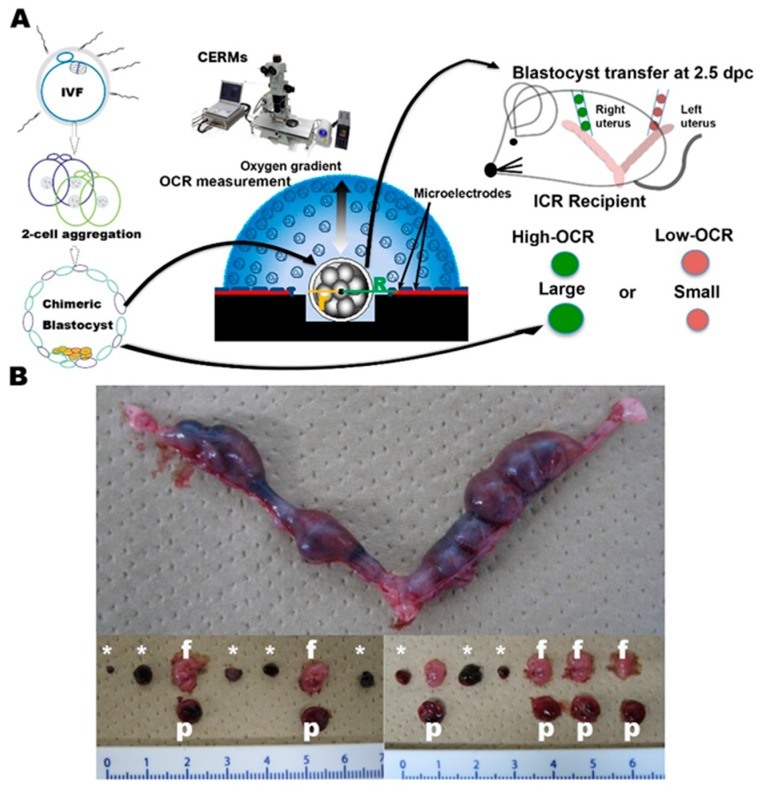
Embryo transfer based on the OCR and the effect of OCR measurement by CERMs on embryo development. (**A**) Schematic diagram representing embryo transfer based on the OCR. Chimera embryos with high- and low-OCR were transferred to each side of a bicornuate uterus of a synchronized-ICR recipient female. (**B**) Pictures show uterus recovered at 12.5 dpc. Numbers of implantation sites and fetal development were assessed. *p*, f, and * indicate placenta, fetus, and implantation site, respectively.

**Table 1 ijms-20-05650-t001:** Relationship between OCR and implantation and viable pregnancy rates.

	Rep Number	Number of Embryos Transferred	Mean OCR (fmol/s)	Mean Diameter (μm)	Implantation Rate (%)	Fetus Rate (%)	Average Fetus Weight (g)
High-OCR	6	60	14.71 ± 4.61 *	113.08 ± 19.27	34/60 (56.6)	10/60 (16.7)	80.93 ± 26.4
Low-OCR	6	60	7.65 ± 1.50 *	110.67 ± 21.28	42/60 (60.0)	23/60 (38.3)	90.37 ± 28.5
			* *p* < 0.05	NS	NS	NS	NS

* Asterisk indicates statistical significance (*p* < 0.05); NS: Not significant.

**Table 2 ijms-20-05650-t002:** Relationship between embryo morphology (size) and implantation and viable pregnancy rates.

	Rep Number	Number of Embryos Transferred	Mean OCR (fmol/s)	Mean Diameter (μm)	Implantation Rate (%)	Fetus Rate (%)	Average Fetus Weight (g)
Large	3	30	9.28 ± 3.01	135.18 ± 13.28 *	20/30 (66.7)	13/30 (43.3)	74.34 ± 20.0
Small	3	30	10.2 ± 3.96	108.40 ± 13.19 *	17/30 (56.7)	8/30 (26.7)	72.10 ± 28.8
			NS	* *p* < 0.05	NS	NS	NS

* Asterisk indicates statistical significance (*p* < 0.05); NS: Not significant.

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
