# Peer review of "A Preclinical Evaluation towards the Clinical Application of Oxygen Consumption Measurement by CERMs by a Mouse Chimera Model"

_ijms, 2019, doi:10.3390/ijms20225650_

Round 1
Reviewer 1 Report
I read with great interest the Manuscript titled “A preclinical evaluation towards the clinical application of oxygen consumption measurement by CERMs by mouse chimera model” (ijms-634248), which falls within the aim of International Journal of Molecular Sciences.
In my honest opinion, the topic is interesting enough to attract the readers’ attention. Methodology is accurate and conclusions are supported by the data analysis. Nevertheless, authors should clarify some points and improve the discussion citing relevant and novel key articles about the topic.
Authors should consider the following recommendations:
Manuscript should be further revised by a native English speaker. The authors have not adequately highlighted the strengths and limitations of their study. I suggest better specifying these points. Embryo transfer is a key stage in IVF, in which the quality of performance determines the outcome. According to recent evidence, transvaginal ultrasound guidance of the transfer significantly increases the percentage of pregnancies per transfer, both in the general population and in the reference population, compared with transfers performed under transabdominal ultrasound guidance. Authors should add few lines about the topic, referring to: PMID: 29576332; PMID: 28428123. Accumulating evidence suggests that approximately 30% of oxygen consumption is due to non-mitochondrial process, which results in reactive species of oxygen (ROS) production. In this regard, I would recommend discussing, at least briefly, the detrimental role of ROS for embryo quality and embryo implantation (refer to: PMID: 29743986; PMID: 27651794).
Author Response
Itemized responses to the reviewers’ suggestions
Note: Revised parts of the manuscript, including both the additions and the deletions are depicted by automatic color annotation of “Track change” function by Microsoft Word or strikethroughs (completely deletions). Revision points where difficult to recognize by automatic color font annotation, we highlighted with yellow. Specification of the points (page and line #) where we revised, were based on the highlighted version of manuscript. The line number may not be consistently applied in order due to “Track change mode” or an incompatibility between Windows and Mac.
We thank the reviewers for precise suggestions for improvement of our manuscript. Our itemized responses to each comment are provided below.
Reviewer #1:
Comment-1
Manuscript should be further revised by a native English speaker. The authors have not adequately highlighted the strengths and limitations of their study. I suggest better specifying these points.
Response: We thoroughly revised the abstract in order to highlight significance and the significance and the limitation of current study in page1, L16-33. Further, we performed professional English proofreading
Comment-2
Embryo transfer is a key stage in IVF, in which the quality of performance determines the outcome. According to recent evidence, transvaginal ultrasound guidance of the transfer significantly increases the percentage of pregnancies per transfer, both in the general population and in the reference population, compared with transfers performed under transabdominal ultrasound guidance. Authors should add few lines about the topic, referring to: PMID: 29576332; PMID: 28428123.
Response: We thank you to great suggestion. We inserted pointed new topics as follows in page 1, line 42-43 to page 2. Line 118-119.
“Although previous efforts have improved ART outcome, such as the change from transabdominal to transvaginal ultrasound guidance during embryo transfer and an innovation of single medium development[4-6] ,the selection of a high-quality embryo for transfer is justified to increase the chance of pregnancy and reduce the risk of miscarriage.”
Comment-3
Accumulating evidence suggests that approximately 30% of oxygen consumption is due to non-mitochondrial process, which results in reactive species of oxygen (ROS) production. In this regard, I would recommend discussing, at least briefly, the detrimental role of ROS for embryo quality and embryo implantation (refer to: PMID: 29743986; PMID: 27651794)..
Response:
We thank you to great suggestion. We inserted suggested new topics and references as follows in page 11, line 688-692.
“Further, the production of ATP via mitochondrial OXPHOS pathway would inevitably increase the production of reactive oxygen species (ROS) production, called mitochondrial ROS (mROS). High level of mROS, a consequence of the overactive mitochondrial OXPHOS, may have a negative impact on cell structures and macromolecules, such as lipids, membranes, proteins, and DNA, and thereby embryo quality[55, 56].”

Reviewer 2 Report
Using mouse blastocysts, authors investigated whether the OCR measured by CERMs correlates with known mitochondrial activity markers, also examining the pregnancy outcomes (stated as “in vivo embryo development or progeny fertility”).
This is a very well conducted study. I have some suggestions in an effort to improve the quality and mainly the smoothness of the reading of the paper.
The abstract, although unstructured, should contain the basic sections that should be referred appropriately.
As an IVF specialist, clinically – and methologically - I would prefer to see primary and secondary endpoints: that is, first authors should be referred to pregnancy outcomes, and then to the previous associations.
The discussion section should have a more structured format. There are plenty of inputs analyzed, but there is no logical and easy to read synthesis. Direct comparisons with the literature should be also employed, together with the limitations of the study and future justified prospects.
Author Response
Itemized responses to the reviewers’ suggestions
Note: Revised parts of the manuscript, including both the additions and the deletions are depicted by automatic color annotation of “Track change” function by Microsoft Word or strikethroughs (completely deletions). Revision points where difficult to recognize by automatic color font annotation, we highlighted with yellow. Specification of the points (page and line #) where we revised, were based on the highlighted version of manuscript. The line number may not be consistently applied in order due to “Track change mode” or an incompatibility between Windows and Mac.
Reviewer #2:
We thank the reviewers for precise suggestions for improvement of our manuscript. Our itemized responses to each comment are provided below.
Comment-1
The abstract, although unstructured, should contain the basic sections that should be referred appropriately.
Response: As responded to the reviewer #1, we thoroughly revised the abstract in order to highlight significance and the significance and the limitation of current study in page1, L16-33. Further, we performed professional English proofreading
Comment-2
As an IVF specialist, clinically – and methologically - I would prefer to see primary and secondary endpoints: that is, first authors should be referred to pregnancy outcomes, and then to the previous associations.
Response:
We agreed to your very logical suggestion and we moved the safety assessment via embryo transfer to the top of results in this study. Therefore, the result 2.1.5, 2.1.1, 2.1.2, 2.1.3 and 2.1.4 are now 2.1.1, 2.1.2, 2.1.3, 2.1.4 and 2.1.5, respectively.
However, since the OCR did not correlate with the ATP production and the cell counts in single blastocysts, we had to initially validate the relationship in the OCR and “known” mitochondrial OXPHOS related biological parameters in chimera model. Chimeric mouse embryos exhibited reasonable correlation in the OCR and ATP, mitochondrial membrane potential and cell numbers, we therefore thought that the OCR value measured by CERMs can be justified as an indirect assessment of oxidative metabolism. We convinced that the mouse chimera model is served as the acceptable animal model for further assessment of CERMs, whereby we moved forward to embryo transfer experiment. Therefore, we would like to keep subsequent order of results from 2.1.2..
Comment-3
The discussion section should have a more structured format. There are plenty of inputs analyzed, but there is no logical and easy to read synthesis. Direct comparisons with the literature should be also employed, together with the limitations of the study and future justified prospects.
Response:
We reconstructed and revised the discussion in accordance with the order change in the results. Changes were as follows;
We moved paragraph start with “Establishing an animal model is indispensable for…” in the original manuscript page10, line 379-392 to the 2nd paragraph in the revise highlighted manuscript in page 8, line 409-416 to page 9, line 434 to 441. We also edited and moved the 4th and 5th paragraphs, start with “Our initial attempts to measure the OCR for…” and “On the other hand, if embryo size (morphology) is a major…”, in the original manuscript page 9, line 320-335 to the 3rd and 4th paragraphs in the revise highlighted manuscript in page 9, line 442-460. We completely removed the 8th paragraph start with “The current embryo evaluation methods may have been developed…” in the original manuscript page 10, line 393-411. Finally, we edited and revised last paragraph to clarify the significances and the limitations of our present study.
Other corrections
We made a minor revision on Figure 5(A) and delated Figure 5(C) due to the order rearrangement of results.
We modified “Abbreviations” and “References” according to the changes made in the main text (page15 line635-page16 line636, page 19 line 756-779).
